# Review on Detection Methods of Nitrogen Species in Air, Soil and Water

**Md Faishal Yousuf** and **Md Shaad Mahmud** *

Department of Electrical and Computer Engineering, University of New Hampshire, Durham, NH 03824, USA;
mdfaishal.yousuf@unh.edu
* Correspondence: mdshaad.mahmud@unh.edu

**Abstract:** Nitrogen species present in the atmosphere, soil, and water play a vital role in ecosystem stability. Reactive nitrogen gases are key air quality indicators and are responsible for atmospheric ozone layer depletion. Soil nitrogen species are one of the primary macronutrients for plant growth. Species of nitrogen in water are essential indicators of water quality, and they play an important role in aquatic environment monitoring. Anthropogenic activities have highly impacted the natural balance of the nitrogen species. Therefore, it is critical to monitor nitrogen concentrations in different environments continuously. Various methods have been explored to measure the concentration of nitrogen species in the air, soil, and water. Here, we review the recent advancements in optical and electrochemical sensing methods for measuring nitrogen concentration in the air, soil, and water. We have discussed the advantages and disadvantages of the existing methods and the future prospects. This will serve as a reference for researchers working with environment pollution and precision agriculture.

**Keywords:** nitrogen; sensor; remote sensing; electrochemical; optical; ammonia; air pollution; monitoring

## 1. Introduction

Nitrogen is the most abundant element in our planet's atmosphere. Nitrogen gas ($N_2$) makes approximately 78 percent of the earth's atmosphere [1]. Chemically, it is an essential component of biomolecules, such as amino acids and nucleotides, and it is the fourth most prevalent chemical in living organisms. The increase in inorganic nitrogen in the environment promotes the growth of living organisms. The world's population is expected to reach roughly nine billion people in the next thirty years, necessitating an increase in food production by more than seventy percent [2]. Farmers use approximately 190 million metric tons of fertilizer in an inefficient and uncontrolled manner in order to address this challenge [3]. The increased use of fertilizers, along with other anthropogenic activities, such as fossil fuel burning and industrialization, has contributed to an increase in reactive nitrogen in the biosphere by a factor of 13 since 1850 [4]. Reactive nitrogen gases in the atmosphere have become a significant cause of air pollution and breathing-related diseases in the human body [5]. When excess nitrogen enters the water, it causes the overgrowth of algae. This algae boom creates a dead zone by absorbing excessive oxygen and blocking the sunlight. According to current estimates of the situation, there are approximately 400 dead zones in the oceans, covering an area four times greater than it was in 1950 [4]. As a result, monitoring the nitrogen species in the environment has become an important topic [6].

Researchers have explored several electrochemical and optical methods for detecting nitrogen species in various environments (Figure 1) [7–17]. Traditional approaches required manual sample collection that then needed to be assessed in the lab, where lab analysis produced precise results. However, lab analysis is not always practical because of the time and resources that it requires. Therefore, the focus shifted to in situ studies. Several attempts have been made for real-time in situ analyses, of which, only a few found their way

into commercial application. This article reviews the recent advances in nitrogen detection techniques and discusses the potential future directions for real-time in situ monitoring.

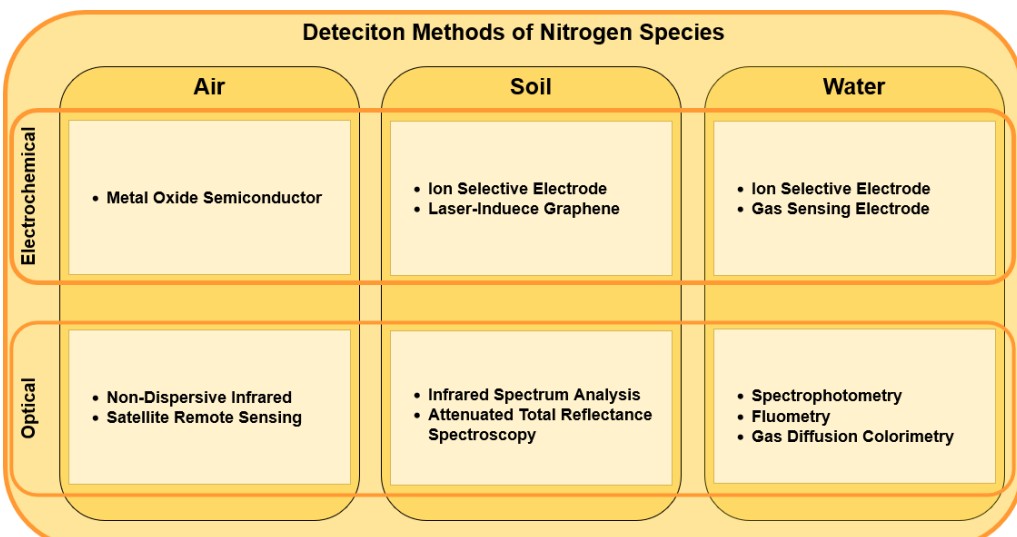

**Figure 1.** Deteciton methods of nitrogen species.

## 2. Species of Nitrogen in Nature

Many biomolecules, such as proteins, DNA, and chlorophyll, require nitrogen as a component. Unfortunately, naturally abundant dinitrogen gas ($N_2$) is chemically inert, which makes it inaccessible to most organisms. As a result, nitrogen has to go through various transformations in its life cycle before the primary producers can absorb it [18]. Nitrogen fixation and ammonification, nitrification, and denitrification are the primary conversion processes. These processes allow for the existence of nitrogen in both organic (e.g., amino and nucleic acids) and inorganic (e.g., ammonia, nitrate) forms. Various microorganisms in the biosphere, such as bacteria, archaea, and fungi, also aid in these changes [19].

The nitrogen cycle is significantly influenced by human activity [20]. The amount of biologically available nitrogen in a system can surge due to burning fossil fuels, applying nitrogen-based fertilizers, and other activities (Figure 2) [21]. Significant changes in nitrogen availability can induce severe adjustments in the nitrogen cycle in aquatic and terrestrial ecosystems because nitrogen availability typically limits the primary productivity of many ecosystems. Studies show that human activity and industrialization have increased nitrogen fixation at an exponential rate since the 1940s [22].

The consequences of excess N are manifold, ranging from the eutrophication of terrestrial and aquatic systems to global acidification and stratospheric ozone loss. Excess nitrogen in the soil is transported away by surface runoff and water moving through the ground. Eventually, it ends up in the water and other ecosystems that can also obtain nitrogen from precipitation. Through these mechanisms, nitrogen enters surface water bodies, altering nature and hastening its aging or eutrophication by boosting the growth of algae and aquatic plants. As these aquatic organisms die and decay, this produces aesthetic concerns and depletes the oxygen content in the water [23].

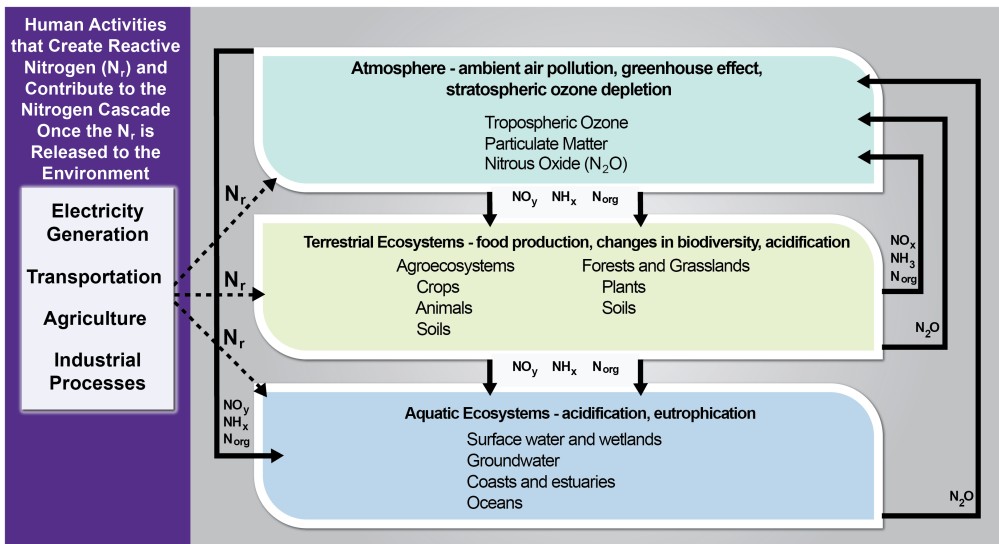

**Figure 2.** Human activities that form reactive nitrogen, and resulting consequences in the environment. [Image credit: Galloway et al., Climate Change Impacts in the United States: The Third National Climate Assessment; Technical Report (U.S. Global Change Research Program; 2014) [24]].

## 3. Nitrogen Species Detection in Air

A large pool of nitrogen gas ($N_2$) exists in the Earth's atmosphere. At normal temperatures, molecular nitrogen ($N_2$) is an inert gas. Other nitrogen gases ($NO_x$) generated by anthropogenic sources, on the other hand, are harmful to the environment. Nitrous oxide ($N_2O$) is one of the significant greenhouse gases responsible for ozone layer depletion; nitric oxide (NO) is a free radical catalyst that is also accountable for ozone layer depletion; nitrogen dioxide ($NO_2$) is a key air pollutant that causes acid rain. Various strategies have been investigated to monitor the ($NO_x$) gases in the air due to these serious concerns.

### 3.1. Electrochemical Sensors for Nitrogen Species Detection in Air
Metal Oxide Semiconductors (MOS)

Metal oxide semiconductors (MOSs) outperform other gas-sensing materials due to their superior physical and chemical capabilities and their distinct structure. MOS-based gas sensors can identify a gas by detecting the change in the electrical signal caused by the gas. Conductometric, impedimetric, and field-effect transistor analyses are three typical characterization approaches for MOS-based gas sensors. Zinc oxide (ZnO), stannic oxide ($SnO_2$), molybdenum trioxide ($MoO_3$), nickel(II) oxide (NiO), and copper(I) oxide ($Cu_2O$) are examples of n-type and p-type MOS gas-sensing materials that are commonly utilized (Figure 3). They are wide-band-gap semiconductors with electrical conductivity that changes in response to the gas composition around them [25]. The semiconductor is split into the gas-interacting surface, the gas-unaffected bulk, and the particle boundary in the middle. The particle boundary is set at a distance equal to the Debye length from any substance exposed to the atmosphere: the space within the sensor across which chemical electrostatic influences can propagate, as indicated by the element's physical characteristics. Oxygen atoms bond to the border at high temperatures, withdrawing electrons from the semiconductor's surface region. After that, the oxygen reacts with the surrounding gases or binds to the sensor, removing or introducing new charge carriers into the surface region [26–30].

MOS sensors are popular for the production of inexpensive gas-sensing devices [31]. The idea of using inexpensive MOS sensors to monitor urban air quality has been popular for some time [28,30]. Even though research on MOS-based gas sensors is growing, the attention dedicated to this mechanism is still insufficient. It has much potential and many problems, such as increasing the gas sensitivity of the material and establishing a comprehensive evaluation system for comparing different mechanisms.

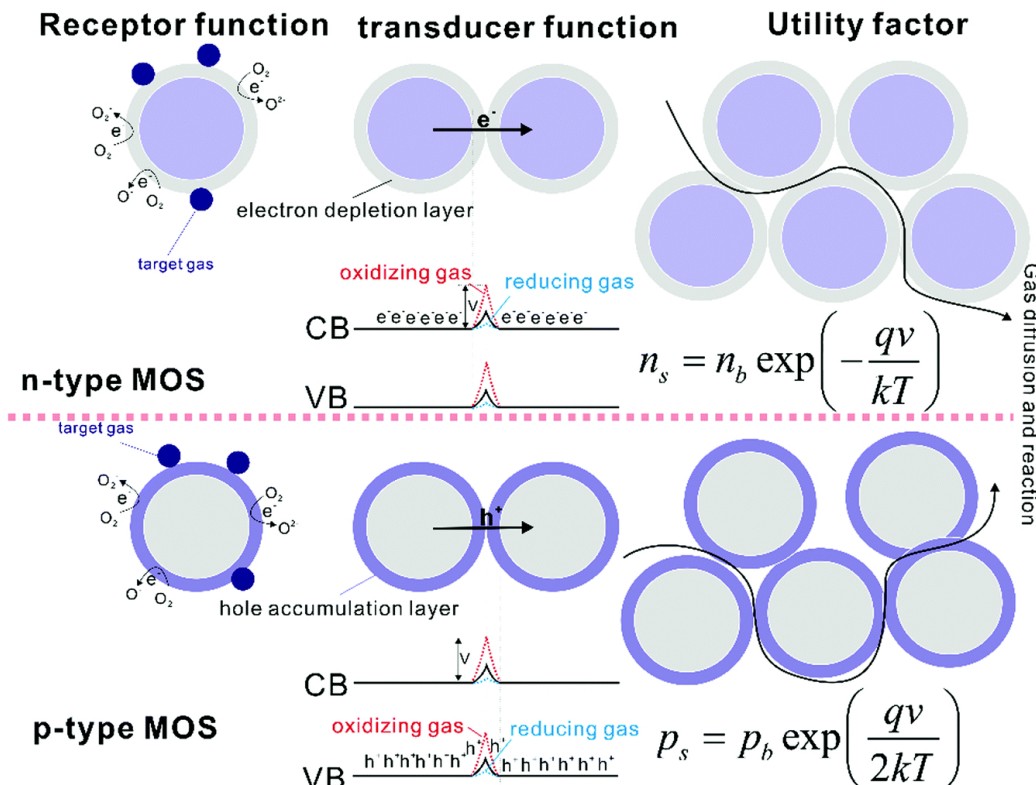

**Figure 3.** The n-type and p-type MOS-based gas-sensing method and conduction model. [Reprinted with permission from ref. [32]].

### *3.2. Optical Sensors for Nitrogen Species Detection in Air*

### 3.2.1. Non-Dispersive InfraRed (NDIR)

The NDIR gas measurement technology looks for the signature wavelength in the infrared spectrum to identify specific gases. A typical NDIR detector (Figure 4) consists of a light source, gas cell (often coated with Au since it is less reactive), band-pass filter, and detector. The band-pass filter is carefully engineered to match an absorption feature of the analyte gas. Since many pollutant gases are IR active, they can potentially be monitored by the NDIR method. Along with other gases, ammonia ($NH_3$), nitrous oxide ($N_2O$), and nitrogen oxides (NOx) have been measured via this method [33].

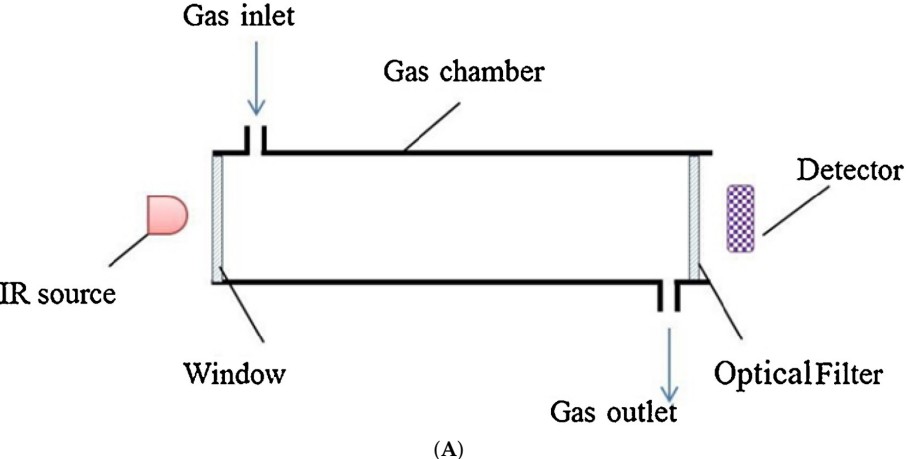

(**A**)

**Figure 4.** *Cont.*

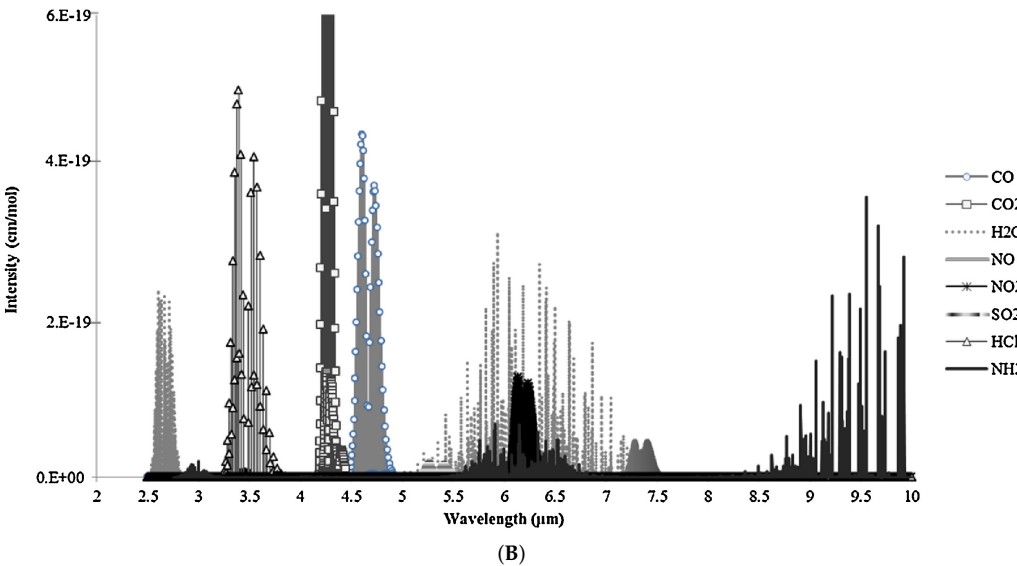

**Figure 4.** (**A**) Schematic of typical NDIR sensor for gas detection. (**B**) Infrared absorption features for several gases. [Reprinted with permission from ref. [34]].

The devices are simple, which is an advantage of NDIR measurements. NDIR sensors can be made tiny and portable, and they only need a small amount of power. The significant disadvantages of the NDIR are interference by other particles and a high limit of detection.

### 3.2.2. Satellite Remote Sensing

Over the last few decades, geospatial data analysis employing satellite imagery has become increasingly common. Several studies [35–41] have reported tropospheric $NO_2$ column observation using satellites to extrapolate ground level NOx concentrations. Satellites equipped with an imaging spectrometer recognize solar emissions returned or disseminated back to space from the Earth's surface or atmosphere. Each target atmospheric gas has a unique spectral fingerprint. The concentration can be measured by recognizing the constituents' unique fingerprints from the electromagnetic spectrum ot the recorded data. Hichem et al. [42] measured $NO_2$ from May 2018 to June 2019 across the territory of mainland France using spectral data collected by the Copernicus Sentinel 5P satellite of the European Space Agency (ESA) (Figure 5). These findings imply that satellite tropospheric $NO_2$ column retrievals and ground-level $NO_2$ concentrations are correlated.

While satellite-derived data on air pollution offer researchers the opportunity to estimate air particles, there are a few drawbacks. First, this is an indirect measuring method suitable for evaluating a large area over a long period. The measurement must be scaled based on local conditions and requires calibration from ground-based monitors for a smaller area [43]. Furthermore, satellites are unable to detect ground-level conditions on foggy days.

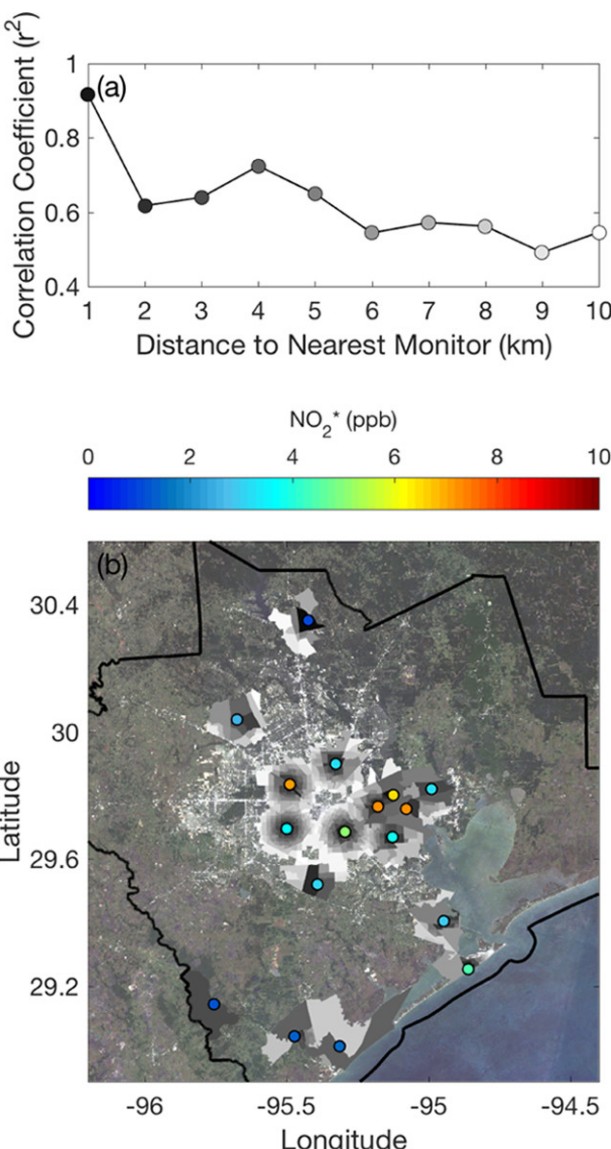

**Figure 5.** NO$_2$ measurement across mainland France using ESA satellite data. (**a**) Correlation coefficient between the overhead columns and the tract-averaged NO$_2$ (**b**) distance between the closest surface NO$_2$ monitoring site and the census tract center point. [Reprinted with permission from ref. [41]].

## 4. Nitrogen Species Detection in Soil

There are two primary sources of nitrogen in the soil that plants may use: nitrogen-containing minerals and atmospheric nitrogen. Nitrogen exists in the atmosphere in a very inert N$_2$ form and must be transformed to other states before being used in the soil. Organic nitrogen molecules, ammonium (NH$_4^+$) ions, and nitrate (NO$_3^-$) ions are the three main types of nitrogen in the soil. A large number of ammonium (NH$_4^+$) ions are also fixed into the soil by added fertilizer. The majority of nitrogen sensing in the soil is dependent on detecting the amounts of these ions.

### 4.1. Electrochemical Sensors for Nitrogen Species Detection in Soil

#### 4.1.1. Ion-Selective Electrodes

Ion-selective electrodes (ISE) are potentiometric sensors for measuring ion activity in a solution. The sensing component of the ISE is an ion-selective membrane (Figure 6). Various membrane-selective soil nutrients (such as nitrate, sodium, potassium, and calcium) have been developed and are commercially available. This type of sensor depends

on the electrolyte function; namely, the electrolyte plays a significant role in the sensor performance. Therefore, most shortcomings arise with a malfunction of electrolytes. For instance, liquid electrolyte-based sensors suffer because a substantial amount of electrolytes is consumed for each detection. The total lifespan of the sensor is reduced as the amount of liquid electrolytes in the sensor decreases. With solid electrolyte-based sensors, it suffers from the possibility of electrolyte poisoning, and its application is limited because of the relatively high required operating temperature. According to the literature, the majority of electrochemical sensors are used to detect pollutants, contagions, and nutrients in aquatic environments [44]. However, ISE's have also been used in soil extracts, moist soils, and slurries for measurements [13].

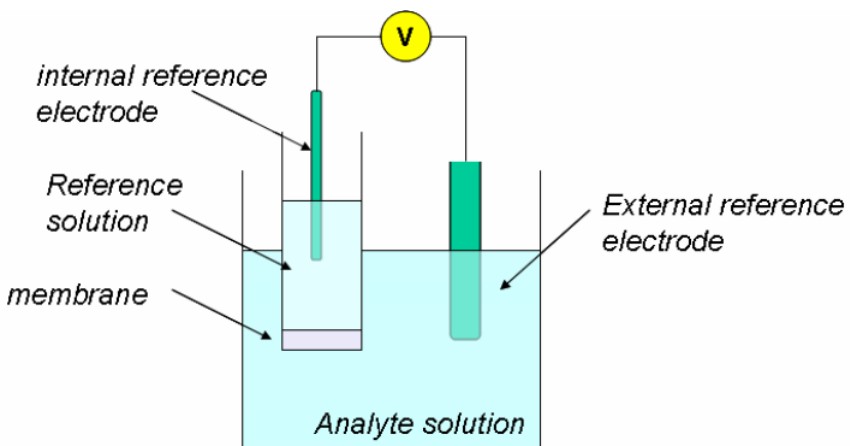

**Figure 6.** Electrochemical cell for a potentiometric measurement with ISE. [Reprinted from ref. [45], by Pavan M. V. Raja & Andrew R. Barron; via OpenStax CNX (CC BY 4.0)].

The measurement irregularities induced by the following drawbacks are the reason for these sensors' limited success. Using electrochemical sensors to evaluate soil nutrients needs a nutrient extraction technique or device, as well as a rinse agent, which adds to the analysis time. Aside from the sensor's need for periodic rinsing, sophisticated calibration requires understanding the soil texture and physical factors. In addition, highly selective membranes must be developed for the accuracy improvement of in situ soil nitrate ISEs to asses the nutrients.

### 4.1.2. Laser-Induced Graphene Sensor

Due to graphene's unique material features, including its high flexibility, electrical and thermal conductivity, and tensile strength, graphene-based electronics hold enormous potential for a wide range of applications. However, it necessitates a lengthy fabrication procedure. Laser-induced graphene (LIG) is a relatively recent alternative to printed graphene circuits that uses a one-step laser writing production approach to generate flexible graphene electronics on polyimide substrates [46].

Garland et al. showed how to employ a low-cost UV laser to make a LIG sensor for the soil samples' ion-selective detection of a plant's nitrogen ions. They created LIG electrodes with different laser pulse widths on polyimide/Epson printer paper and found electrodes with a pulse width of 20 ms to be the most effective. The LIG electrode was then used with an ionophore membrane that is selective for $NH_4^+$ or $NO_3^-$ to develop the ISE (Figure 7).

This research demonstrates that LIG-based electrodes are equivalent to the recent trend of screen printing low-cost carbon-based electrodes. However, the process is significantly more straightforward than graphene-printing approaches. LIG electrodes are appropriate for use in disposable sensor technologies and can be manufactured and scaled roll-to-roll. Sensor calibration, however, is still required.

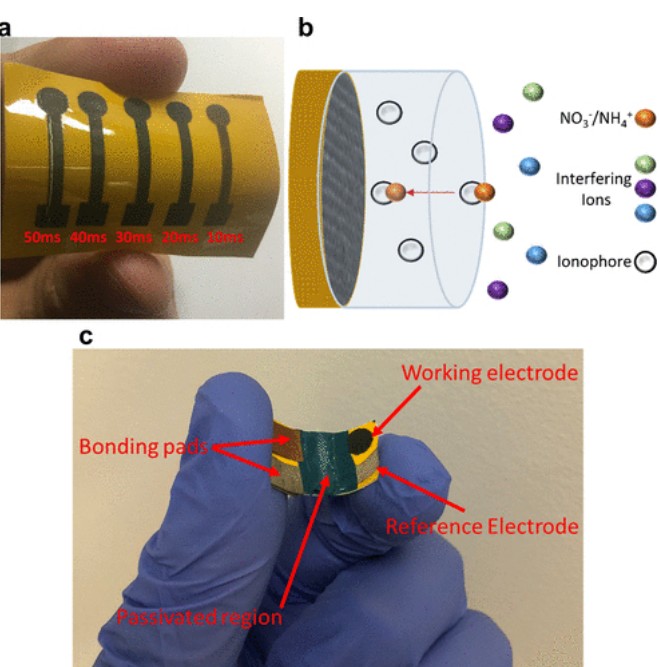

**Figure 7.** (**a**) Photograph of five LIG SC ISEs on a single polyimide swatch. (**b**) Illustration of SC-ISE ion sensing. (**c**) Representative electrode used in soil column studies. Passivated regions are shown as well as bonding pads, working electrode, and reference electrode. [Reprinted with permission from ref. [46]].

*4.2. Optical Sensors for Nitrogen Species Detection in Soil*

4.2.1. Infrared Spectrum Analysis (IRS)

Infrared spectroscopy is gaining popularity due to the speedy and cost-effective prediction of soil's physical, organic, and chemical properties [47–51]. The most common spectra utilized in the spectroscopic investigation of soil and water contents are ultraviolet (UV) [52], near-infrared (NIR) [14], and mid-infrared (MIR) [53]. Joose et al. [54] conducted a comparative analysis of various IR ranges and discovered that the best predictions for physical and chemical qualities were generally found in the MIR spectral range; they also stated that the optimal spectral range selection depends on the soil property. Jose et al. experimented with both VIS, Vis-NIR and Vis-MIR for soil properties analysis [54], and could predict the total nitrogen among other components. Nie et al. used NIR spectroscopy soil in a penetrating probe for measuring soil nitrogen [55]. However these measurements were performed on treated soil samples in a lab environment.

4.2.2. Attenuated Total Reflectance Spectroscopy

Attenuated total reflectance (ATR) as a spectroscopic sample presentation technique has been widely used in many areas, including food, agriculture, and environmental science. The basic principle is the same as infrared spectroscopy. However, rather than lighting the sample and receiving diffused reflectance spectra, the infrared energy is directed into a crystal set that is directly in contact with the target sample that has a higher refractive index. (Figure 8). As a result, the incident energy is reflected multiple times within the crystal, resulting in an evanescent field at the sample–crystal contact. When the internally reflected energy eventually leaves the system, it is sent to the spectrophotometer, which generates the sample's reflected spectrum. Then, nitrogen contents, along with other soil components, can be estimated from the spectrum.

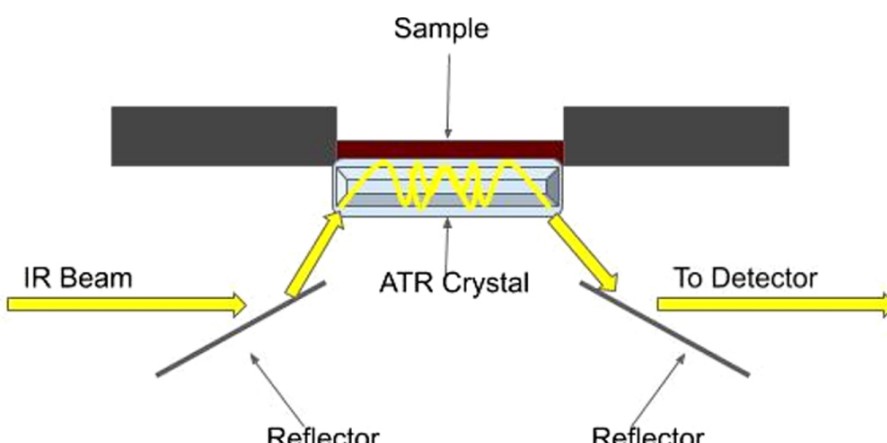

**Figure 8.** ATR spectroscopy illustration. When an infrared beam enters the ATR device, it is reflected by a series of reflectors and directed toward the ATR crystal that is in contact with target sample. The light is reflected by the crystal producing an evanescence. After leaving the ATR crystal, the reflected energy is directed toward a spectrometer, which generates a sample reflectance spectrum. [Reprinted from ref. [13], by Lamar Burton1, K. Jayachandran2 and S. Bhansali; via IOP (CC BY 4.0)].

Shao et al. [56] used Fourier transform infrared attenuated total reflection spectroscopy (FTIR-ATR) in a lab environment and reported that the method could well predict the nitrate concentration. Many studies [57–59] reported that direct analysis of samples using ATR spectroscopy requires minimal sample pretreatment. However, the presence of soil moisture and other minerals causes interferences, which requires advanced data processing. In addition, this method is not suitable for in situ analysis because of the cost of the devices involved.

## 5. Nitrogen Species Detection in Water

Water nitrogen comprises un-ionized ammonia ($NH_3$) and ammonium ($NH_4^+$) ions that are affected by pH and temperature. In most aquatic systems, ammonia nitrogen is mostly present as ammonium, but when the pH and temperature rise, the fraction of un-ionized ammonia increases [60]. $NH_3$ and $NH_4^+$ together contribute to the total ammonia nitrogen concentration of water, which can be measured in various ways.

### 5.1. Electrochemical Sensors for Nitrogen Species Detection in Water

#### 5.1.1. Ion-Selective Electrodes

Ion detection with ion-selective electrodes (ISE) is a well-established technique. The main working principle of ISE is already discussed in the soil nitrogen detection section. In most natural water, nitrogen mainly exists as an ammonium ion. The polyvinyl chloride (PVC) membrane is the most widely used ammonium ion-selective membrane. However, there is a continuous search for a better and more sensitive electrode. Liquid-contact membranes are affected by the reference liquid, temperature, and pH of the solution [61]. The advancement in solid-contact ISE has provided a new direction for in situ environmental water analysis [62,63]. Being easy to use and having a quick response time, compactness, low energy consumption, low production costs, and a high dynamic response range are just a few of the advantages of in situ analysis using ISE [64].

#### 5.1.2. Gas Sensing Electrodes

The gas-sensing electrode features a hydrophilic permeable membrane that separates ammonia from the aqueous sample using an internal ammonium chloride solution (Figure 9) [8]. The ammonium salt is transformed into ammonia before being injected into the internal liquid through a gas-sensitive membrane that filters ammonia to flow through. The concentration is then determined by measuring the potential signal. This method is comparatively common and mature. However, recent research has focused on using auxiliary procedures to minimize

measurement needs or to improve the measurement accuracy. However, this method has to prove successful in detecting low levels of ammonia nitrogen [65–67].

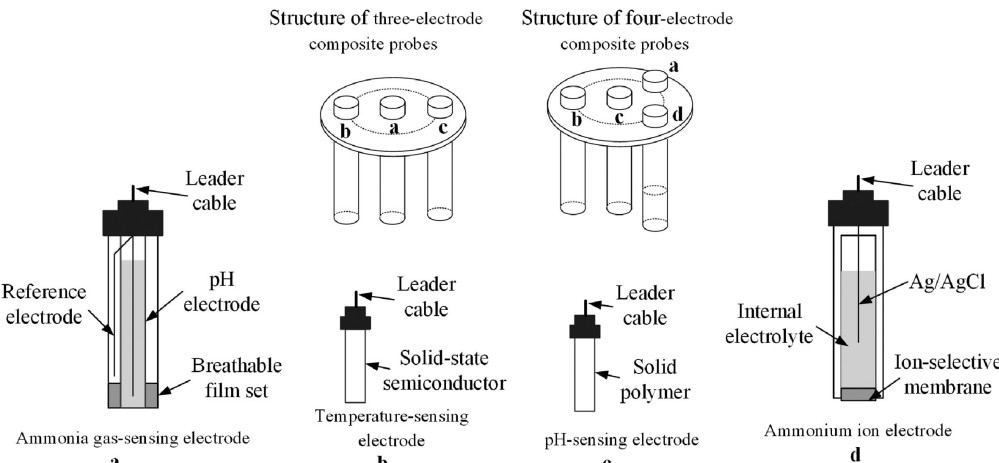

**Figure 9.** Type and structure of multi-parameter composite probes. [Reprinted with permission from ref. [65]].

*5.2. Optical Sensors for Nitrogen Species Detection in Water*

5.2.1. Spectrophotometry

Nessler's reagent is a long-established reagent for determining ammonia in water. When ammonia combines with Nessler's reagent, it produces a yellow solution. Nxumalo et al. [68] and Phansi et al. [69] studied the efficiency of the microfluidic paper-based device ($\mu$PAD) for assessing ammonia in wastewater using Nessler's reagent approach in recent years. Phansi et al. used a smartphone camera to identify color intensities in the solution after the reaction, whereas Nxumalo et al. used a digital camera. In another study, Bao et al. [70] employed a computer camera to detect nitrogen concentrations using Nessler's reagent. The spectrophotometric detection method, which uses Nessler's reagent, is comparatively more straightforward, although it is not without drawbacks. Nessler's reagent is a poisonous chemical with a short shelf life (around three weeks). Calcium, magnesium, and other ions can also cause problems with the approach. The pH of the solution is a critical element in the quantification of ammonia, and it warrants further examination [68].

Over the past years, the Berthelot reaction-based indophenol blue (IPB) method has been most commonly used to determine ammonia nitrogen in sea water. Reagents used in the classic IPB approach include phenol, hypochlorite, and nitroprusside [71]. In alkaline circumstances, ammonia reacts with hypochlorite to form monochloramine, which subsequently reacts with two phenol molecules to form indophenol, a blue-colored compound. The intensity of the color change directly relates to the concentration of the ammonia [72].

The Berthelot reaction-based IPB approach offers a good detection range and detection limit, allowing for on-line portable detection and application to a wide range of real water samples. The conventional IPB method's main drawback is reagent toxicity [73], which can be avoided by utilizing ortho-phenylphenol (OPP) [73] or salicylic acid and its salts instead of poisonous phenol. The salicylate technique was shown to be more suitable for use in aquaculture in a recent study [74], since it produces no dangerous secondary pollution and has excellent precision and accuracy. Figure 10 shows ammonia concentration at different pH for Nessler's reagen and IPB mehtod.

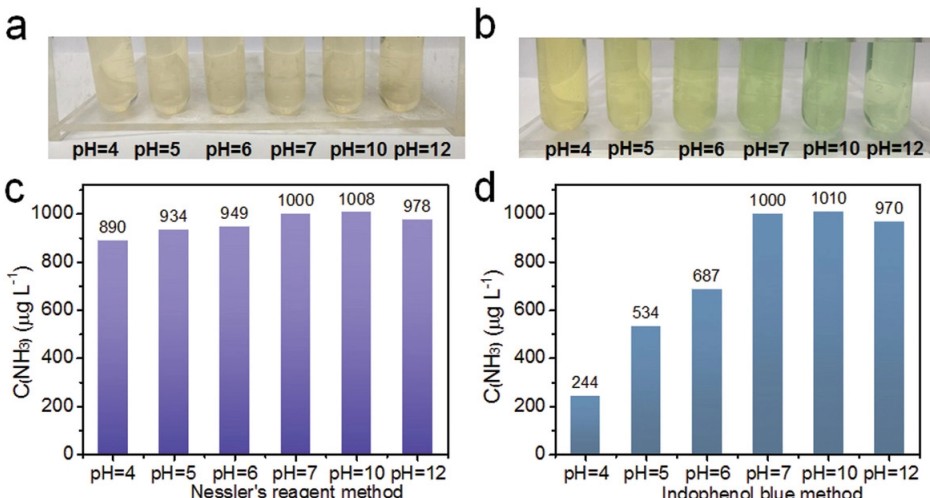

**Figure 10.** Ammonia concentration detection using (**a**,**c**) Nessler's reagent and (**b**,**d**) indophenol blue at different pH. [Reprinted with permission from ref. [75]].

### 5.2.2. Fluometry

A fluorescence compound absorbs light energy at one wavelength and re-emits it at a longer wavelength almost instantaneously. In recent years, several researchers have used this method to determine ammonia nitrogen in aquatic systems. A fluorescent compound is formed when ammonia nitrogen reacts with o-phthalaldehyde (OPA) and sodium sulfite in an alkaline medium. This compound emits light of a particular wavelength and intensity when external energy is applied to it. The intensity of the light is proportional to the nitrogen content [65]. This method was initially developed for detecting amino acids. Over the years, researchers have used, modified, and improved this method to detect ammonia nitrogen by addressing the interference, sensitivity, and sample separation issues [71]. The reaction of NH3-o-phthalaldehyde(OPA)-sulfite is affected by the pH. At a pH > 10.4, precipitation occurs due to metal ions in the natural water samples. Hu et al. [76] used ethylenediaminetetraacetate-NaOH (EDTA-NaOH) as a buffer during the reaction, which achieved a greater sensitivity for quantifying ultra-trace ammonium. Liang et al. [77] reported a novel fluorescent reagent, 4-methoxyphthalaldehyde (MOPA), that produces fluorescent substances reacting with ammonium at room temperature. Zhang et al. [78] synthesized another novel fluorescent reagent, 4,5-dimethoxyphthalaldehyde, for the in situ determination of ammonium. They showed that laser diodes (LDs) as light sources for the laser-induced fluorescence (LIF) detector could perform acceptable detection.

### 5.2.3. Gas Diffusion Colorimetry

Colorimetric gas sensors are a promising low-cost, low-power option for identifying gases in the ambient air quickly and easily. The detection process is based on tracking a color change in a gas-sensitive dye that, in theory, responds selectively to only one target gas by changing color [79]. This allows researchers to determine ammonium in water by converting it to ammonia in a gas diffusion unit under alkaline conditions. First, converted ammonia diffuses to an acid-base indicator solution, causing a color change in the indicator. The absorbance is then measured using a spectrophotometer. Bromothymol blue is commonly used as the indicator [80]. Among other indicators, Sukaram et al. [81] used a natural indicator extracted from the orchid flower to detect ammonia in wastewater, changing its color from purple to green.

Gas diffusion colorimetric approaches have recently shown a lot of potential for use with paper-based analytical equipment (μPAD) [55,82]. The gas separation mechanism of this method lowers interference and delivers a higher and more consistent detection accuracy.

## 6. Discussion and Future Prospects

Nitrogen species are present in both organic and inorganic form and travel from one environment to another through the nitrogen cycle. Researchers have explored various methods to monitor this flow effectively. Most of the attention is directed to electrochemical and optical detection methods. Both methods have advantages and disadvantages depending on the target species and the detection environment.

The detection of nitrogen species in the atmosphere has traditionally focused on ammonia gas and nitrogen oxides because of these gases' toxicity. The use of metal oxide gas sensors to detect gaseous nitrogen species has received a great deal of attention and application. However, metal oxide gas sensors operate at a very high temperature, which is an obstacle to real-time detection and energy efficiency [83]. Carbon materials, such as graphene, have been explored to address the temperature issue. Though the application of carbon materials lowers the temperature, more research is needed to make it operational at the average air temperature. Alternatively, spectroscopic optical methods are highly effective at a wide range of temperatures. In addition, spectroscopic methods have a longer life and provide quick analysis, making them ideal for commercial applications. Laser absorption and NDIR spectroscopy have been found to be effective in detecting gaseous nitrogen species where the effective wavelength range is 1450 to 1560 nm in the spectrum. Shao et al. [84] developed a laser sensor emitting at a 4.54 µm frequency, using tunable diode laser absorption spectroscopy (TDLAS), that could detect $N_2O$ and CO simultaneously. However, future focus should be given to addressing the issue of interference by vapor and other particles. Other optical sensing methods, such as satellite data analysis, can estimate nitrogen species for a certain period over a large area. However, they are not suitable for real-time monitoring due to the nature of the data collection process and the postprocessing of the data collected by the satellites.

For soil nitrogen species detection, ion-selective electrode sensors are the most popular electrochemical sensing sensor. However, these sensors work well in moist soil or require soil pretreatment for better accuracy. In addition, ISEs are very selective to their target and not suitable for multiple species detection at a time. The fundamental role of electrochemical sensors is manipulating the reaction at the interface of the electrode and solution to utilize the electron transfer as an analytical signal. Hence, surface modification plays a vital role in developing a high-performance electrochemical sensor. In addition, they work best in a controlled environment, such as a laboratory. Their use in the field is limited since they are delicate and expensive, have a short lifespan, require frequent calibration, and are susceptible to variables, such as the pH of the sample [85]. On the other hand, soil spectroscopy does not require any chemical reagent since it does not involve any chemical reaction. This method has shown high potential in both the laboratory and field applications. Recent advances in portable spectrometer development have made this method suitable for in situ monitoring. Fillipe et al. [85] developed a modular compact sensing system for soil NPK using direct UV-Vis spectroscopy combined with an optical fiber. They showed that an analytical grade quantification of NPK is possible by recording absorption spectra and applying a self-learning artificial intelligence algorithm to them. However, it is necessary to carry out more research to overcome the interference from soil moisture and other minerals. In addition, the type of the soil plays an important role in spectral analysis since the absorbance or reflectance of light varies depending on soil texture. Therefore, further study is required to see how spectroscopic methods perform based on the soil type.

Optical sensors have been used in agriculture, industry, healthcare systems, environmental monitoring, and space science for decades. They have undergone tremendous growth and advancement in recent years. As a result of their dielectric properties, they can be employed in high-voltage, high-temperature, and corrosive situations. Furthermore, these sensors are suitable for interfacing with data communication systems and remote sensing. Optical sensing is the ideal answer for applications where traditional electrical sen-

sors have proven ineffectual or challenging to utilize, such as extreme climatic conditions and detecting across great distances [86].

Globally, farmers are looking for advanced precision technologies to assist them in changing their agri-tech practices into more sustainable and productive ones. As a result, soil data that are accurate and updated in real-time have become one of the most significant resources for farmers. Precision agriculture uses remotely sensed data to help farmers reduce their resource input while increasing their yield [13]. Real-time remote sensing is also crucial for environmental sensing to prevent air and water pollution. Up to this point, different techniques have been employed in a variety of settings. There is no single method that can be applied in multiple environments due to the differences in each environment. Whereas electrochemical sensors are highly selective, optical sensors can open the door for multivariate sensing. The application of optical spectroscopy in this situation has great potential. Spectroscopic methods have been applied in air, soil, and water environments (Table 1). Since the chemical structures have a unique signature in spectra at a different wavelength, it would be exciting to look further into it. Optical sensors of different wavelengths can be coupled to capture the spectral response at a wide range of wavelengths. Then, modern machine learning algorithms can be used to analyze the data. This can open a new door for in situ analysis. Modern optical sensors can be coupled with machine learning and AI to produce results comparable to laboratory analysis. A gradual decrease in size and cost of optical sensors, increase in longevity, and advancement in edge AI technology have made it of great importance to explore these areas. An array of these sensors can be deployed to collect localized data and develop statistical models by combining and validating with data from other sources, such as satellite data, to map a whole region [87–89].

**Table 1.** Optical sensing methods in various environments.

| Environment | Sensing Method | Detected Components | References |
|---|---|---|---|
| Soil | IRS Spectroscopy | Soil Nitrogen | [47–50] |
| Soil | ATR Spectroscopy | Nitrate | [56] |
| Water | ATR Spectroscopy | Nitrate | [15,90] |
| Water | UV Spectroscopy | Nitrate | [52] |
| Air | FTIR Spectroscopy | $NO_2$ | [91] |
| Air | NDIR | $NO_2$ | [33] |
| Air | TDLAS | $NO_2$ | [84] |
| Air | LED Absorption Spectroscopy | $NO_2$ | [92] |

## 7. Conclusions

Various sensing techniques for nitrate species detection in diverse settings have been widely investigated in recent times. Despite the fact that other approaches have been investigated, electrochemical sensors have received the most interest for their sensitivity and simplicity. However, due to surface fouling and their low stability, they must be maintained and calibrated on a regular basis. As a result, to meet environmental and agricultural needs, we need robust in situ sensors that can give long-term real-time analyses. Optical sensors have recently gained popularity among researchers due to their compact size, longer lifetime, immunity to diverse interference, and lower cost. In addition, there has been a minimal attempt to replace traditional mathematical calibration and sensor data processing models with cutting-edge machine learning techniques. With recent advances in edge computing and edge AI, it is clear that experimenting with and implementing these new technologies can result in more accurate, long-lasting real-time nitrogen sensing.

**Author Contributions:** M.F.Y. conducted most of the experiments, data curation and writing. M.S.M. was responsible for draft preparation and supervision of this project. All authors have read and agreed to the published version of the manuscript.

**Funding:** This work was supported by NSF SitS Award: 1920965.

**Data Availability Statement:** Not Applicable.

**Conflicts of Interest:** The authors declare that the research was conducted in the absence of any commercial or financial relationships that could be construed as a potential conflict of interest.

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
