# Peer review of "Review on Detection Methods of Nitrogen Species in Air, Soil and Water"

_nitrogen, doi:10.3390/nitrogen3010008_

Round 1
Reviewer 1 Report
The review paper is interesting and provide important and valuable information regarding Nitrogen measurements in several environments.
The authors have improved considerably the manuscript.
The abstract flows well and the ideas are better structured.
In the introduction, the authors are making the case for the paper in a nice way.
The topics 2 to 5 presents well the concepts and Figures 1 to 10 are very good.
Line 83: check “Mo3” What it means?
Line 379: The phrase “In this study, various nitrogen sensing approaches are discussed.” Is not a good way to start a conclusion of a manuscript. Please consider improve it.
Author Response
1. Line 83: check “Mo3” What it means? -> This is Molybdenum trioxide (MoO3) whose chemical formula was put wrongly. We have corrected the formula. 2. Line 379: The phrase “In this study, various nitrogen sensing approaches are discussed.” Is not a good way to start a conclusion of a manuscript. Please consider improve it. -> Thank you for the suggestion. We have changed the wording to improve the sentence.Reviewer 2 Report
The comments from my original review have been fully addressed in the revised manuscript. Thus, I recommend publication in its present form.
Author Response
Thank you very much for your support!
Reviewer 3 Report
This manuscript presents review of detection methods of nitrogen species in air, soil and water. Authors review the recent advancements in optical and electrochemical sensing methods for measuring nitrogen concentration. The authors carefully looked at the current state of literature. The number of references is satisfactory
The article can be accepted after minor correction for publication in Nitrogen.
line 142 - ISE - The abbreviation is introduced here but the explanation is on line 218
Author Response
Thank you very much!
1. line 142 - ISE - The abbreviation is introduced here but the explanation is on line 218 -> We have added the abbreviation on the first use of the full form.